

# Epidemiological research on parent–child conflict in the United States: subgroup variations by place of birth and ethnicity, 2002–2013

Jose Ruben Parra-Cardona[1], Hsueh-Han Yeh[2] and James C. Anthony[2]

[1] Human Development and Family Studies, Michigan State University, East Lansing, MI, United States
[2] Epidemiology and Biostatistics, Michigan State University, East Lansing, MI, United States

Corresponding author
Jose Ruben Parra-Cardona,
parracar@msu.edu

## ABSTRACT

**Background**. Chronically escalated parent–child conflict has been observed to elicit maladaptive behavior and reduced psychological well-being in children and youth. In this epidemiological study, we sought to estimate the occurrence of escalated parent–child conflict for United States (US) adolescent subgroups defined by (a) ethnic self-identification, and (b) nativity (US-born versus foreign-born).

**Methods**. US study populations of 12-to-17-year-olds were sampled, recruited, and assessed for the National Surveys on Drug Use and Health (NSDUH), 2002–2013 ($n = 111,129$). Analysis-weighted contingency table analyses contrasted US-born versus foreign-born who self-identified as: (a) Hispanic, (b) non-Hispanic African-American, (c) non-Hispanic Asian, and (c) non-Hispanic White.

**Results**. Frequently escalated parent–child conflict was most prevalent among US-born non-Hispanic White adolescents, from 18% at age 12 (95% CI [17.6%, 18.9%]) to 29% at age 17 (95% CI [28.3%, 29.7%]), followed by US-born Hispanic and non-Hispanic Asian children. Estimated prevalence proportions were markedly lower for African-American children, from 8% at age 12 (95% CI [6.8, 8.5]) to 16% at age 17 (95% CI [14.3, 16.7]). Broad and sometimes overlapping CI indicate that larger sample sizes are needed for complete evaluation of an apparent excess occurrence of frequent parent-child conflict among US-born versus foreign-born. Nonetheless, in the larger subgroups, the US-born show a clear excess occurrence of frequent parent-child conflict. For example, US-born Mexican children have 1.7 times higher odds of experiencing frequent parent-child conflict than foreign-born Mexican children (OR = 1.7, 95% CI [1.5, 2.0], *p*-value < 0.001).

**Discussion**. The main discovery from this multi-ethnic sample investigation is a rank-ordering of parent-child conflict prevalence estimates from high (non-Hispanic White) to low (non-Hispanic African-American). The pattern also suggests a possibly generalizable excess associated with US-born sub-groups. The epidemiological estimates presented here merit attention in future cross-cultural research focused on parent-child conflict.

## INTRODUCTION

In epidemiology, there is a long tradition of research on disease rates before and after migration from one country to another, as well as rural–urban migration within a country. Generally, the intent has been to estimate the degree to which cultural and social environmental processes (e.g., change in diet) might affect general and mental health, as well as successful adaptation, maladaptation, longevity, and case fatality rates (*Syme, 1971*). For example, many epidemiological studies with immigrant populations have examined the role of diet, lifestyle, and culture as etiological determinants of heart disease (*Holmboe-Ottesen & Wandel, 2012*; *Yano et al., 1979*). These themes continue to be prominent in contemporary public health research and epidemiological field studies that integrate cultural, social, and interpersonal influences on health-related behaviors, as well as investigations of how social capital might change or differ across migrant groups (*Miranda et al., 2011*; *Velderman et al., 2015*; *Alarcón et al., 2016*). Recently, *Acevedo-Garcia et al. (2012)* proposed a cross-national framework for the study of immigrant health, and incorporated constructs from epidemiology with those of economics and other social sciences. The present research inquiry, based on recent large sample epidemiological surveys conducted in the United States (US), was designed as a contribution to this tradition of research on health and adaptation of foreign-born immigrants as compared to homeland-born peers. By design, the size and diversity of the US epidemiological field survey samples have given the study an interesting capacity to make contrasts of the foreign-born versus the US-born within a limited number of sub-population strata defined by ethnic self-identification and by age.

The relevance of the current study can be situated at the intersection of epidemiology and public health research. Specifically, within the field of public health, a pair of inter-related mental hygiene and child guidance movements of the early 20th century combined to foster multiple lines of family studies on effective parenting. Resulting evidence draws attention to supportive and nurturing parent–child relationships and their influences on successful adolescent development, well-being, and academic success, with reduced risk for child internalizing and externalizing disturbances (*Bjorknes & Manger, 2013*; *Dishion & Kavanagh, 2003*; *Kaminski et al., 2008*; *Lundahl, Nimer & Parsons, 2006*; *Seedall & Anthony, 2013*).

### Parent–child conflict

At normative levels, parent–child conflict seems to foster successful adaptations, an increased definition of self, and essential life skills (e.g., negotiation with authority), among other important developmental milestones (*Fuligni, 2012*; *Moed et al., 2015*). When parent–child conflict is inappropriately managed, outcomes can include escalating conflict and hostility, sometimes concurrent with maladaptive adolescent behavior and reduced psychological well-being (*Bradford, Vaughn & Barber, 2008*; *Patterson, Reid & Eddy, 2002*; *Timmons & Margolin, 2015*).

The study of parent–child conflict has been historically situated in the fields of sociology, psychology, and human development (*Laursen, Coy & Collins, 1998*; *Updegraff et al., 2012*). Prominent among these lines of investigation are studies focused on understanding the influence of parent–child conflict on the development of maladaptive behavior in children

and youth. Among existing theoretical frameworks, the coercion model has been identified as highly influential (*Forgatch et al., 2009*; *Forgatch & Domenech Rodríguez, 2016*).

The origins of the coercion model can be traced back to mid-1960s research focused on understanding origins of persistent child aggression and antisocial behavior (*Dishion et al., 2016*). Whereas many parent–child conflict studies had been sociological in nature, the Oregon group developed research more focused on behavior analysis of observed parent–child interactions in an effort to increase research objectivity (*Dishion et al., 2016*). One result was the coercion model, according to which harsh and ineffective parenting can shape a child's risk for antisocial behaviors and development of socially maladaptative behavior (*Wachlarowicz et al., 2012*).

In essence, the coercion model postulates that "ineffective parenting and deviant peer association are the prime mechanisms of changes in forms of deviancy" (*Forgatch et al., 2009*, p. 640). Central to the coercion model are ineffective parenting practices, increasing occurrence of parent–child conflict interactions characterized by negative reciprocity (i.e., mutual criticisms), aversive behaviors (e.g., yelling, negative comments) and escalation of conflict (i.e., arguments leading to increased verbal exchanges). The coercion model postulates that children are at greater risk for developing maladaptive behaviors if they are frequently exposed to parenting that includes frequent parent–child conflict.

Multiple prevention and clinical intervention studies have confirmed utility of the coercion model as a framework for understanding and improving deleterious parent–child interactions and for improvement of protective parenting practices (*Dishion & Andrews, 1995*; *Dishion et al., 2016*). This study is focused on one specific element of the coercion model—namely, frequent parent–child conflict.

## Cross-cultural parent-child conflict

Origins of the idea that ethnicity, nativity, and immigration status might affect parent–child conflict and later mental health or behavior can be traced back to early anthropological and sociological studies of culture, personality, social structure, and migration as summarized by *Mead (1947)* and *Merton (1949)*, as well as more recent theoretical formulations and empirical evidence on topics such as 'acculturation stress' experienced by adolescents and parents in immigrant families (e.g., see *Sluzki, 1979*; *Acevedo-Garcia et al., 2012*; *Warner & Swisher, 2015*; *Alarcón et al., 2016*; *Szapocznik, Kurtines & Fernandez, 1980*). This background of evidence set the stage for the current contribution of new epidemiological evidence on parent–child conflict as viewed from the adolescent's perspective.

The current investigation responds to a call for research on cross-cultural similarities and differences of parent–child interactions (*Dmitrieva et al., 2004*). Multi-ethnic studies in this domain may be particularly relevant when parent–child interactions such as frequent conflict are shaped by differences across cultures or ethnicities (*Uji et al., 2014*; *Updegraff et al., 2012*). For example, recent studies contrasting Asian families suggest multi-ethnic variations in the combination of strict parental directives and control with a warm and nurturing relationship (*Lau & Fung, 2013*; *Tran & Birman, 2010*).

Empirical estimates from the present study draw strength from recently completed epidemiological field surveys with nationally representative samples of community
populations within the United States (US), primarily intended to illuminate child and adult drug involvement and health. In aggregate, these surveys produced samples large enough for statistically precise estimates of parent–child conflict among adolescents distinguished by ethnic self-identification and to some extent by place of birth ('nativity'). A review of previously published parent–child conflict studies shows very few estimates from epidemiologically and nationally representative samples of ethnically diverse families (e.g., see *Moed et al., 2015*; *Cain & Combs-Orme, 2005*; *Hughes et al., 2006*; *Juang & Umana-Taylor, 2012*). Partially as a reflection of sample availability, the opportunities for between-group comparisons have been limited (*Fuligni, 2012*; *Juang & Umana-Taylor, 2012*; *Chung, Flook & Fuligni, 2009*).

### Growing interest in public health research focused on multi-ethnic samples

Growing global interest in public health research has prompted researchers to call for investigations of cross-cultural differences and similarities in developmental processes, with attention to fundamental structures of parent-adolescent relationships, family dynamics, and parenting practices that might vary according to ethnicity (*Acevedo-Garcia et al., 2012*; *Dmitrieva et al., 2004*). A focus on multi-ethnic samples within the United States can be motivated without difficulty when one appreciates recent dramatic changes and increased diversity in the US population structure and composition (*United States Census Bureau, 2012*).

In addition, multi-ethnic studies with a focus on parent–child conflict have importance to the extent that traditional conceptions of parent–child relationships are subject to cross-cultural influences (*Dmitrieva et al., 2004*; *Updegraff et al., 2012*). For example, in many US studies to date, emphasis has been given to 'acculturation stress' or 'acculturation gap-distress' hypotheses, according to which cultural differences between immigrant parents and children can create or modulate parent–child conflict (*Lau et al., 2005*; *Updegraff et al., 2012*). Partial support for these hypothesized relationships can be seen, but recent studies illustrate a need to refine basic conceptualizations (*Updegraff et al., 2012*). That is, possibilities for parent–child conflict might be enhanced for immigrant families of one ethnic subgroup, but might be dampened for other ethnic subgroups (*Shearman & Dumlao, 2008*; *Yu & Singh, 2012*).

The quality of the parent–child conflict data in our current public health research on families is quite limited and cannot be used to settle debates about the acculturation gap-distress hypothesis or any other complex cross-cultural theory that might account for observed relative frequency of parent–child conflict. In keeping with the tradition of public health surveillance operations in general, epidemiological surveys are designed for practicality and brevity. The survey assessment modules provided detailed coverage of drug use, but faced tight constraints on assessments of other topics such as parenting practices. Nevertheless, we discovered that the survey's multi-ethnic samples were large enough to shed light on a single basic facet of parent–child conflict as it is distributed across mutually exclusive ethnic subgroups, with exploration of age-specific contrasts between

12-to-17-year-olds born in the US versus the foreign-born, and with statistical adjustments for expected male–female differences.

## Purpose of the study

Against this rich background, this study's main aim was to estimate occurrence of frequent parent–child conflict within a nationally representative and ethnically diverse sample of 12-17 year olds residing in US communities. Of special interest was variation in frequent parent–child conflict in relation to child ethnic self-identification (i.e., non-Hispanic African American, non-Hispanic Asian, Hispanic, non-Hispanic White) and country of origin (i.e., US vs. foreign born). The resulting estimates are especially relevant due to the nationally representative and multi-ethnic group sampling design. Prior parent–child conflict studies of this type have not been epidemiological in nature and have had a limited representation of ethnic minority groups (*Moed et al., 2015*; *Juang & Umana-Taylor, 2012*). Thus, this investigation is situated at the under-studied intersection of epidemiology and public health research with research on parent–child interactions pertinent to child health and development.

Before turning to a description of materials and methods, we should clarify that the 'frequent parent–child conflict' construct, as studied here, should not be assumed to signify 'inept parenting' or any consequence of inept parenting. Instead, frequent parent–child conflict of this type might be regarded as a 'surveillance signal' of potential public health importance. In this respect, the study's estimates for frequent parent–child conflict might serve well as population norms when US family counselors, pediatricians, and other clinicians seek to compare an individual patient's parent–child conflict frequency with a population reference group of similar age, ethnic background, and immigration status. We return to these issues in our Discussion section.

With regards to study hypotheses, we thought that we might see more frequent parent–child conflict in the subgroup of non-Hispanic White adolescents, irrespective of US-born or foreign-born status, perhaps traceable to greater value assigned to child autonomy in Euro-American cultures. As for children self-identifying as Hispanic or as Asian, we anticipated lower parent–child conflict values, due to what others have characterized as a greater respect for parental authority and hierarchy in these subgroups. Nevertheless, we also wondered about whether there might be more parent–child conflict among the US-born Hispanic and Asian adolescents, as they might bid for greater autonomy, as compared to their peers in the foreign-born first generation immigrant adolescent subgroups. Finally, racial socialization literature on parenting practices in African American families informed our expectation for lower levels of parent–child conflict for these adolescents, associated with promotion of respect to family structure and parental authority.

## METHOD

### Research design, probability sampling, and sample size

In this investigation, the cross sectional survey design was set up to yield nationally representative estimates for the US federal government's program known as the National Surveys on Drug Use and Health (NSDUH), 2002–2013, with study populations

encompassing all non-institutionalized US civilian community residents age 12 years and older during the survey years. That is, each year's study population included 12-to-17-year-olds in non-institutional community dwelling units, irrespective of whether they were attending school (i.e., with no 'school survey' restriction to children still in school). The survey population is defined to exclude relatively small segments of the current US non-institutionalized population of 12-to-17-year-olds such as homeless children who do not reside in shelters and children living on military bases. In addition, child residents of institutional group quarters such as long-stay psychiatric hospitals also are excluded. Participation in the NSDUH is not restricted to drug users or to individuals with psychiatric or behavioral disorders. Because respondents are not asked about their US citizenship nor immigration naturalization status, it is not possible to bring those US Census variables into play.

Each year's samples were drawn using multi-stage area probability sampling in all 50 states and the District of Columbia. Parental consent and child assent was obtained via protocols approved by cognizant institutional review boards for human subjects protection. In addition, approval for this investigation was obtained from the Michigan State University IRB.

The federal government has made available a large subset of the survey data and variables in the form of a public use dataset known as the NSDUH Restricted-Use Data Analysis System (RDAS). Major advantages of the RDAS public use datasets are (1) detailed variables on US- versus foreign-born status, and (2) pre-constructed analysis weights for aggregate analyses of the pooled 2002–2013 NSDUH survey data. The only exclusions were based on age (with parent–child conflict assessed only for 12-17-year-olds) and non-missing values for age, sex, US-born versus foreign-born, and ethnic self-identification. The result is a large sample size for each of the four subgroups under study, with enhancement of statistical precision and external validity of the study estimates. The effective sample size for this investigation ($n = 111,129$ 12-to-17-year-olds) was determined by the Vsevolozhskaya-Anthony method (*Vsevolozhskaya & Anthony, 2014*). Table 1 shows adolescent subgroup cell counts for the study estimates.

## Assessments

Participants chose either an English or Spanish language version of the NSDUH audio computerized self-interview (ACASI) or a paper-and-pencil option. The parent–child conflict assessment was in a module positioned midway through the 60–90 min NSDUH session, with the following introduction: "…*the next question asks about your parents. By parents, we mean either your biological parents, adoptive parents, stepparents, or adult guardians who live in your household.*" The parent–child conflict item was: "*During the past 12 months, how many times have you argued or had a fight with at least one of your parents?*" with an ordered gradient of responses from 0 times, 1–2 times, 3–5 times, 6–9 times, and 10 or more times during the past 12 months. For this surveillance report, we characterized the '10 or more times' response as 'frequent parent–child conflict.' A disadvantage of the RDAS public use dataset is that the use of any general or generalized linear model is not an option for analysis; RDAS permits only frequency distributions,

**Table 1  Characterization of US-born and foreign-born 12-to-17-year-olds in the nationally representative sample.** Approximate unweighted numbers for self-identified ethnic groups (racial/ethnic heritage) by male–female and US/foreign-born origin. Data from the Restricted-use Data Analysis System (RDAS), National Surveys on Drug Use & Health, United States 2002–2013.[a]

| | US-Born ($n = 102,379$) | | Foreign-Born ($n = 8,750$) | |
|---|---|---|---|---|
| | Estimated number and percentages with frequent parent–child conflict | | | |
| | Yes ($n = 23,795$)(%) | No ($n = 78,584$)(%) | Yes ($n = 1,397$)(%) | No ($n = 7,353$)(%) |
| **Non-Hispanic White** | 17,625 (74.06) | 50,690 (64.50) | 518 (37.07) | 2,495 (33.92) |
| **Non-Hispanic African American**[b] | 2,100 (8.83) | 12,583 (16.01) | 67 (4.80) | 405 (5.51) |
| **Hispanic:** | | | | |
| Mexico origin | 2,070 (8.70) | 8,438 (10.74) | 270 (19.33) | 1,860 (25.30) |
| Non-Mexican origin | 1,413 (5.94) | 4,574 (5.82) | 269 (19.26) | 1,339 (18.21) |
| **Non-Hispanic Asian:** | | | | |
| China origin | 175 (0.74) | 621 (0.79) | 64 (4.58) | 205 (2.79) |
| Non-Chinese origin | 412 (1.73) | 1,678 (2.14) | 209 (14.96) | 1,049 (14.27) |

**Notes.**

[a]The RDAS system does not disclose actual cell counts, but *Vsevolozhskaya & Anthony (2014)* provide a method that can be used to derive unweighted effective sample size values as shown here.

[b]Within-subgroup analyses were not calculated due to small sample sizes (e.g., Non-Hispanic African Americans of Caribbean origin vs. African origin).

bivariate tables, and multi-way contingency table analyses. For this reason, we had to choose one threshold value and were unable to model the responses using a generalized linear model for discrete ordered responses.

## Statistical analysis

A standard "explore, analyze, explore" cycle started with Tukey-style exploratory analyses that shed light on underlying distributions, followed by an initial RDAS analysis/estimation step to produce analysis-weighted multi-way contingency tables, from which graphical display was generated. RDAS variance estimation uses the Taylor series 'delta' method (*Lohr, 2009*).

Our attempt to produce sex-specific estimates was thwarted by RDAS non-disclosure constraints described by Vsevolozhskaya and Anthony, but we were able to use the Vsevolozhskaya-Anthony method to derive effective sample sizes for each cell of a series of sex-specific multiway contingency tables based on the following variables: place of nativity (US-born versus foreign-born), sex (male versus female), units of age (from 12 through 17) (*Vsevolozhskaya & Anthony, 2014*). We then fashioned Cochran-Mantel-Haenszel analyses to estimate the degree to which being US-born might be associated with greater odds of frequent parent–child conflict (as compared to the foreign-born reference category), with sex and age held constant (*Cochran, 1954*; *Mantel & Haenszel, 1959*). For this analysis/estimation step, it was possible to derive statistically adjusted odds ratio estimators for the US-born versus foreign-born, specific to: (1) Hispanic adolescents of Mexican heritage, (2) Hispanic adolescents of non-Mexican heritage, (3) Asian adolescents of Chinese heritage, (4) Asian adolescents of non-Chinese heritage, (5) non-Hispanic Black/African-American adolescents, and (6) non-Hispanic White adolescents, as shown in Table 2.

**Table 2   Estimated covariate-adjusted odds ratio linking US-born status with odds of frequency of parent–child conflict, by self-identified ethnic groups.** Data from the United States National Surveys on Drug Use and Health, 2002–2013.[a]

|  | AOR (95% CI)[b] | *p*-value |
|---|---|---|
| **Non-Hispanic White:** | | |
| All NHW children | 1.7 (1.5, 1.9) | <0.001 |
| **Hispanic:** | | |
| Mexican heritage | 1.7 (1.5, 2.0) | <0.001 |
| Non-Mexican heritage | 1.6 (1.4, 1.8) | <0.001 |
| **Non-Hispanic Asian:** | | |
| Chinese heritage | 0.9 (0.6, 1.2) | 0.708 |
| Non-Chinese heritage | 1.3 (1.1, 1.5) | 0.011 |
| **Non-Hispanic African-American** | | |
| All NHAA children | 0.8 (0.7, 1.1) | 0.229 |

**Notes.**
[a] Description of the study sample is provided in the Methods section.
[b] Adjusted odds ratio, AOR, with foreign-born as reference subgroup (covariates = age in years, sex: male versus female).
CI, Confidence interval.

# RESULTS

Before application of the analysis weights to the sample, the sample size for this study encompassed 111,129 12-to-17 year-old survey respondents. An estimated 92% of the participants self-identified as US-born versus 8% foreign-born. Among US-born participants, 67% self-identified as non-Hispanic White, 14% as non-Hispanic African American, 16% as Hispanic, and 2.8% as non-Hispanic Asian. Among the foreign-born participants, 34% self-identified as non-Hispanic White, 5.4% as non-Hispanic African American, 43% as Hispanic, and 17.5% as non-Hispanic Asian.

Overall, foreign-born adolescents were somewhat older than the US-born (e.g., 39% of foreign-born age 16–17 versus 33% of US-born). For both US-born and foreign-born adolescents, almost all lived with the mother (∼90%), and about 75% lived with the father, with no appreciable nativity difference in terms of single-parent versus dual-parent families (data not shown in a table; available upon request).

Figure 1 displays age-specific and ethnicity-specific estimates for the US-born 12-to-17-year-olds, showing a general tendency of an increase, age stratum by age stratum, in the proportion with 'frequent parent–child conflict' (i.e., 10 or more times in the past 12 months), and with clear separation of the US-born non-Hispanic White children versus US-born non-Hispanic African-American children, as well as non-Hispanic Asian children. In contrasts with the larger estimates seen for US-born non-Hispanic Whites (from 18% at age 12 to 29% at age 17), the Hispanic and Asian subgroups are intermediate, although at age 14 and at age 16, the non-Hispanic White and Asian children do not differ appreciably. Moreover, studied age by age, the US-born Hispanic children and the US-born non-Hispanic Asian children do not differ appreciably. Except for the US-born in the earliest years of adolescence, it is the non-Hispanic African Americans who are always observed with the smallest estimates for prevalence of frequent parent–child conflict

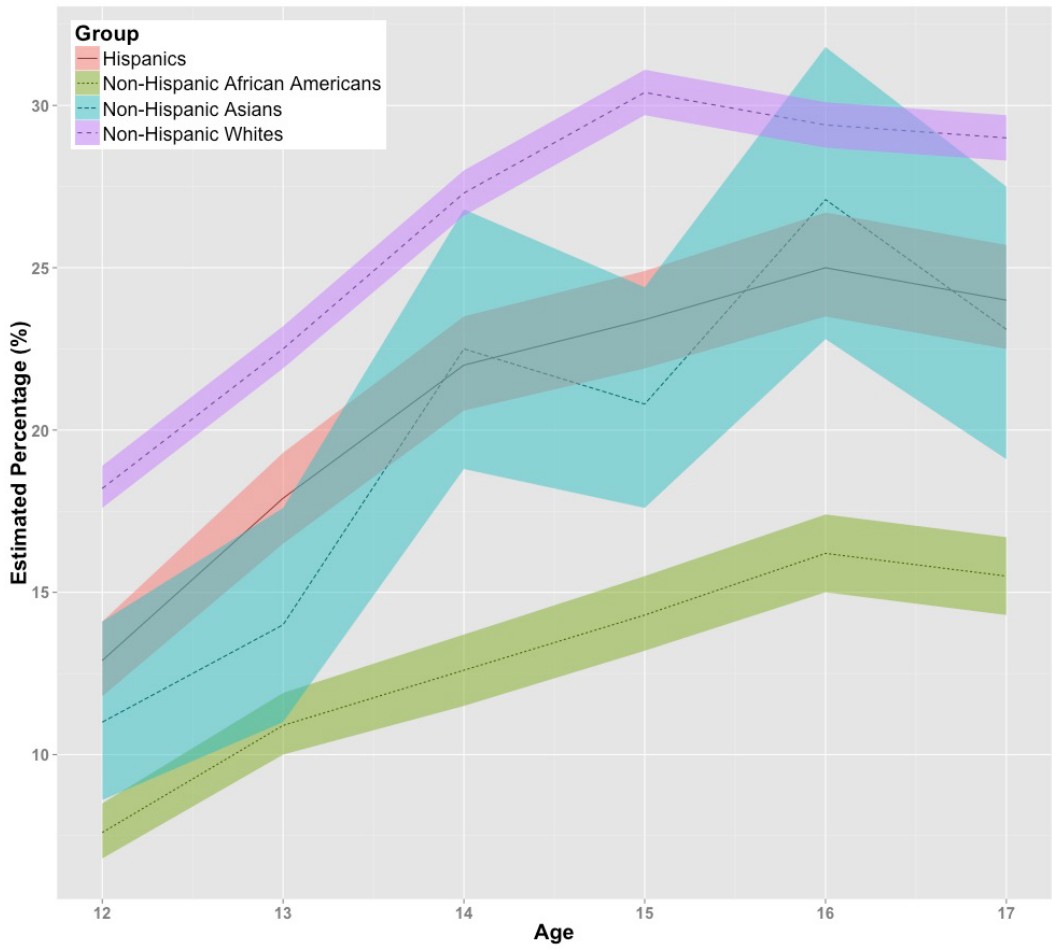

**Figure 1  Estimated percentage with frequent parent-child conflict among US-born 12-to-17-year-olds, with 95% confidence intervals (color-shaded).** Data from the United States National Surveys on Drug Use and Health, 2002–2013 (non-Hispanic subgroups except as noted).

(from 8% at age 12 to 16% at age 17). Except as noted above, among the US-born, the non-Hispanic Asian child and the Hispanic child clearly have intermediate estimates, and the non-Hispanic White children are most likely to have frequent parent–child conflict.

As shown in Fig. 2 and as might be expected with much smaller sample sizes for the foreign-born 12-to-17-year-olds spread over six age values and four subgroups, there is considerable overlap of the 95% confidence intervals. Nevertheless, in these estimates for foreign-born children, it is possible to see a tendency for increases in the proportions with frequent parent–child conflict, age stratum by age stratum, as well as a general preservation of the rank ordering of top and bottom point estimates seen in Fig. 1 for Whites and African-Americans of non-Hispanic heritage. Among the foreign-born, the non-Hispanic Asian children and Hispanic children generally display point estimates of intermediate rank.

The male-to-female ratio for US-born adolescents is 1.04 and is 1.06 for foreign-born adolescents, based on the NSDUH study estimates, and we observed some age differences when contrasting nativity status, as described in this section's first paragraph. Accordingly,

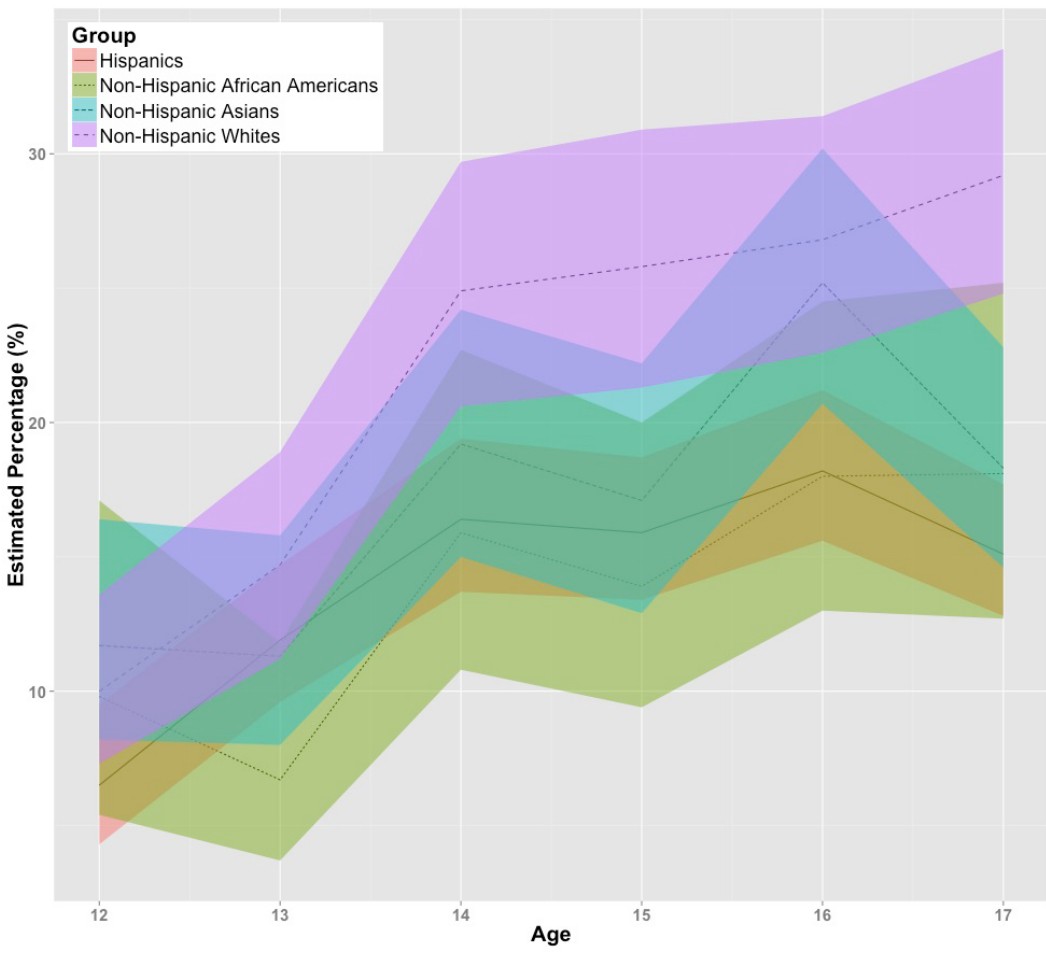

**Figure 2** **Estimated percentage with frequent parent-child conflict among foreign-born 12-to-17-year-olds, with 95% confidence intervals (color-shaded).** Data from the United States National Surveys on Drug Use and Health, 2002–2013 (non-Hispanic subgroups except as noted).

we turned the Cochran-Mantel-Haenszel approach to derive age-, and sex-adjusted odds ratio estimates of the study associations, with an expectation that the odds of frequent parent–child conflict might be greater for the US-born across each of the subgroups under study. The resulting CMH odds ratio estimates are presented in Table 2, starting with a clearly excess odds of frequent parent–child conflict observed for US-born non-Hispanic White 12-to-17-year-olds relative to the foreign-born (odds ratio, OR = 1.7; 95% confidence interval, CI [1.5, 1.9]; Table 2).

Hispanic children with self-identified Mexican heritage were the largest subgroup among the Hispanics in the NSDUH sample. In the contrast of US-born versus foreign-born in this subgroup, the odds ratio estimate was 1.7 (95% CI [1.5, 2.0]; Table 2). As for the US-born foreign-born odds ratio contrast for children of non-Mexican heritage, the US-born again had greater odds of frequent parent–child conflict (OR = 1.6; 95% CI [1.4, 1.8]). It is noteworthy that these first three OR estimates are not appreciably different from one another and the 95% CI overlap considerably.

The corresponding covariate-adjusted odds ratio estimate for US-born versus foreign-born among non-Hispanic Asian child of Chinese heritage is null (OR = 0.9; 95% CI [0.6, 1.2]; $p = 0.708$). However, among non-Hispanic child of non-Chinese heritage, a non-null association is observed (OR = 1.3; 95% CI [1.1, 1.5]; Table 2).

There were too few foreign-born non-Hispanic African-American children to form separate strata for place of birth (e.g., Africa, Caribbean region, South America). When the subgroup of non-Hispanic African-American adolescents is considered in aggregate, the odds of frequent parent–child conflict did not vary across US-born versus foreign-born status (OR = 0.8; 95% CI [0.7, 1.1]; $p = 0.229$).

## DISCUSSION

The current study constitutes a contribution to the existing knowledge base of epidemiological and public health studies focused on migration, with a special focus on differences in family functioning according to ethnic and nativity variations. The main findings of this investigation may be summarized succinctly. First, viewed cross-sectionally among US-born children, there is a general age-associated increase in the presence of frequent parent–child conflict, age stratum by age stratum, irrespective of ethnic self-identification. In general, the subgroup of US-born non-Hispanic White 12-to-17-year-olds has the largest prevalence estimates for frequent parent–child conflict. Relative to this subgroup, the US-born non-Hispanic African-American adolescents have generally lower prevalence of frequent parent–child conflict; the US-born Hispanic and non-Hispanic Asian subgroups had intermediate prevalence estimates. Second, these same general relationships were seen in age-specific estimates for foreign-born children, but there was considerable overlap of confidence intervals. Third, with covariate adjustments for sex and age based on the CMH approach, the US-born in each multi-ethnic subgroup generally were found to have greater odds of frequent parent–child conflict, as compared to the foreign-born, with two exceptions among non-Hispanic children: (1) the Asian children of Chinese heritage, and (2) the African-American children.

Before any detailed discussion of these findings, a review of important strengths and limitations is needed. Strengths include the nationally representative multi-ethnic samples, as large or larger than others have studied, as well as the standardized ACASI assessment approach. Based on the cross-sectional nature of the investigation, the study estimates for foreign-born children invite questions about whether the parent–child conflict originated before immigration or afterward, but this is an issue best resolved in future longitudinal research that can use this study's estimates to set Bayesian priors for what might be expected if and when the cross-sectional research can be extended longitudinally. The assessment of frequent parent–child conflict also is a limitation that can be improved upon in future research, perhaps by asking both the adolescents and their parents about the nature and frequency of parent–child conflicts, as well as the extent to which distressing emotions are associated with such interactions. We cannot mount a strong counter-argument to those who might claim that the observed differences are an artifact of this study's use of an assessment approach based strictly upon the adolescent report. Finally, it is possible

that some readers would prefer a balancing of the NSDUH sample and analysis-weighted distributions to the US Census counts and distributions for US native-born or naturalized citizens, permanent residents, and others. However, the NSDUH does not draw its sample with respect to these 'citizenship' variables, nor does it ask respondents whether they are native-born or naturalized citizens or non-citizen residents of US communities.

We can note that our original plan was to produce estimates for additional US-born versus foreign-born subgroups, but the large NSDUH sample sizes were insufficient to the task. An illustration involves our intent to contrast US-born African-American children versus those born overseas in the non-US Caribbean versus Africa and other places (e.g., Brazil). This intent was undermined when the initial study estimates disclosed age-specific variations too large to ignore. Furthermore, odds ratio estimates for ethnic sub-groups were limited to comparing the largest represented sub-group for each category against remaining sub-groups (e.g., Mexican-origin vs Non Mexican-origin). This approach was needed due to insufficient sample size across all sub-group categories. Notwithstanding an array of study limitations, the study produced novel discoveries that can be pursued in future investigations.

## Place of birth and parent-child conflict

The level of frequent parent–child conflict identified for US-born children is of interest and may reflect cultural differences of a type described previously, including a description by *Szapocznik & Kurtines (1993)*, who were the first to hypothesize this origin for parent–child conflict in US Hispanic families, particularly among Hispanic immigrant families with US-born children. That is, the non-US-born parents of US-born children may experience relationship difficulties due to contrasting levels of acculturation, preferred cultural values, traditions, and ethnic identifications (*Schwartz et al., 2010*; *Harwood et al., 2002*). This phenomenon was originally termed as the "acculturation gap distress hypothesis" based on clinical research with Hispanic immigrant families from Cuba (*Szapocznik & Kurtines, 1993*). Whereas the hypothesis has been confirmed in cross-sectional studies (*Félix-Ortiz, Fernadez & Newcomb, 1998*; *Unger et al., 2009*), additional empirical studies have suggested a need for refinements when the task is to understand or predict parent–child conflict and other youth outcomes such as drug use (*Elder et al., 2005*; *Lau et al., 2005*; Martinez, 2006; *Smokowski, Rose & Bacallao, 2008*; *Alarcón et al., 2016*).

## Parent-child conflict across ethnic groups

Current findings indicate a relatively low occurrence of high-frequency parent–child conflict among non-Hispanic African-American 12-to-17-year-olds, with no marked US-born versus foreign-born variations. This finding deserves more focused inquiry, especially because it tends to call into question deficit-based perspectives about this subgroup's family dynamics (*McAdoo, 2002*).

One consequence of more detailed investigation might be the disclosure of assets in these families that often are neglected in the popular press. These assets already have been covered in prior scholarly evaluation of racial socialization theories about African-American parenting practices—including a need to prepare children for success in a racially stratified

society, with complements in traditional values of respect for parental authority and family structure (*Burke, 2008*; *Burton et al., 2010*; *Peters, 1985*). Here, prior contributions have noted racial socialization practices associated with increased family cohesion, deference to parental authority, and positive child outcomes (*Elmore & Gaylord-Harden, 2013*). Extension of these lines of research in a multi-ethnic society creates opportunities for study of contextual influences that can include imbalanced frequencies of official contacts with US police, prosecutors, judiciary, and criminal justice facilities, as well as general and mental health disparities (*Elmore & Gaylord-Harden, 2013*; *Tamis-LeMonda et al., 2008*; *Mooradian, 2010*; *Burgess et al., 2007*; *Washington et al., 2015*).

With respect to non-Hispanic Asian adolescents, we note a possibility that these families might tend to adopt a parent–child conflict resolution style characterized by a "vertical in-group orientation" in which compliance or avoidance are used to resolve family conflicts. As such, whether US-born or foreign-born, parent–child conflict might increase in intensity but not frequency among Asian immigrant families when there are strong parental expectations of high academic achievement and respect (*Hwang, 2000*; *Qin et al., 2012*; *Costigan & Dokis, 2006*).

As noted above, some recent studies have drawn attention to the combination of rigorous discipline and control in Asian families with warmth and nurturing parent–child emotional bonds (e.g., see *Tran & Birman, 2010*). *Lau & Fung (2013)* characterize the parenting approach in these Asian families as one that offers "a holistic attention to the children's social, moral, and personality development." Here, we confess our disappointment that the aggregate NSDUH samples were not large enough to study variations in frequent parent–child conflict across sub-groups of Asian-heritage families. This deficiency of the current investigation can be remedied as more NSDUH samples become available for study.

As for Hispanic adolescents, this study's observations are generally consistent with prior empirical findings and confirmation of Hispanic parent preferences for parenting practices conducive to family harmony and parent–child bonding (*Domenech Rodriguez, Donovick & Crowley, 2009*). The mid-range occurrence of high-frequency parent–child conflict observed in this study's Hispanic families also might signify prominent Latino cultural values with emphasis upon close and nurturing family relationships (i.e., *familismo*), respect (i.e., *respeto*), and the importance of interpersonal relationships (i.e., *personalismo*) (*Falicov, 2014*). These values may also indicate the children' reluctance to engage in more open communication and disagreement with their parents. For these reasons, more research is needed if we are to develop a more complete understanding of conflict management in US families of Hispanic heritage (*Cookston et al., 2012*), particularly when cultural and contextual influences are prominent (*Calzada et al., 2012*; *Romero, Gonzalez & Smith, 2015*; *Kempf-Leonard, 2007*; *Walker et al., 2004*). In the work of Pantin, Schwartz, and colleagues, with samples of Hispanic youths, a number of acculturation challenges with Parent–youth cultural disagreements have been noted, with potential maladaptive behaviors as a result (*Pantin et al., 2003*; *Schwartz et al., 2012*). Exact mechanisms linking these challenges and disagreements with maladaptation in Hispanic immigrant families remain unclear (*Pasch et al., 2006*; *Unger et al., 2009*). In addition, context may matter, as suggested in a

recent study that found different parent–child acculturation issues when Hispanic families were contrasted across two metropolitan areas of the US (Miami versus Los Angeles, in *Schwartz et al., 2012*). Here, again, capacity of this investigation to shed light on sub-groups within the overall 'Hispanic' category was limited. More epidemiological research on these within-Hispanic category variations should increase the value of the empirical estimates for eventual application in prevention and clinical interventions targeted to Hispanic populations in the US (*Bornstein, 2012*).

The marked excess odds of high frequency parent–child conflict in non-Hispanic White families might reflect tolerance of disagreement within these families, relative to expectations about parenting styles and child autonomy (*Barber, 1994*). One preferred authoritative parenting style in this subgroup places stress on parental guidance, in balance with considerations for the child's individuality and autonomy (*Bornstein, Putnick & Lansford, 2011*). As such, normative processes of parent–child conflict can reflect the adolescent's greater freedom of expression and dissent, as compared with what is seen for adolescents of other ethnic groups, but current investigations fall short in many respects. For example, as Moed and colleagues (*2015*) have noted when examining patterns of interaction of Parent–youth conflict, "how long the conflicts were and who eventually ended them were characteristics of conflicts most linked with the resolution of conflicts and with adolescent problem behavior" (p. 1617). These processes have been neglected in the current epidemiological survey work, but might help explain our estimates for non-Hispanic Whites.

## CONCLUSION

To sum up, we observed interesting patterns of parent–child conflict in these initial nationally representative multi-ethnic sample depictions of one facet of parent–child conflict in the 21st century United States, with clear evidence of US-born versus foreign-born imbalances in several subgroups of interest. At this time, we choose to delay a discussion of specific implications of this research for clinical or public health practice due to the methodological limitations of the study. However, we do note that future extensions of this line of research, building from these initial findings, may be important in prevention and intervention practices that are responsive to emerging cultural shifts within US multi-ethnic subgroups, family structures, and dynamics (*Bornstein, 2010*; *Bornstein, 2012*; *Kagitcibasi, 2013*). In addition, as noted by *Sluzki (1979)*, pediatricians and other clinicians who work with US-born or foreign-born immigrant adolescents in child guidance or pediatric contexts may find this study's nationally representative age-specific study estimates to be useful as population norms when they observe frequent parent–child conflict in their patients or client-families. That is, the results presented here should prove to be useful to United States clinicians who wish to compare an individual patient's parent–child conflict frequency with a population reference group, and to US-based researchers making plans for new research in which frequent parent–child conflict is under study.

### Funding

The study is supported by funds from the National Institute on Drug Abuse grants K01DA036747 (to JRPC), K05DA015799 and T32DA021129 (to JCA), as well as Michigan State University. The funders had no role in study design, data collection and analysis, decision to publish, or preparation of the manuscript.

### Grant Disclosures

The following grant information was disclosed by the authors:
National Institute on Drug Abuse grants: K01DA036747, K05DA015799, T32DA021129.
Michigan State University.

### Competing Interests

The authors declare there are no competing interests.

### Author Contributions

- Jose Ruben Parra-Cardona conceived and designed the experiments, performed the experiments, contributed reagents/materials/analysis tools, wrote the paper, reviewed drafts of the paper.
- Hsueh-Han Yeh conceived and designed the experiments, performed the experiments, analyzed the data, contributed reagents/materials/analysis tools, wrote the paper, prepared figures and/or tables, reviewed drafts of the paper.
- James C. Anthony conceived and designed the experiments, contributed reagents/materials/analysis tools, wrote the paper, reviewed drafts of the paper.

### Human Ethics

The following information was supplied relating to ethical approvals (i.e., approving body and any reference numbers):

Data for this study was administered by the US federal government's program known as the National Surveys on Drug Use and Health (NSDUH). The link below provides direct access to all the information regarding human subjects approval, including consent process, associated with these surveys: https://nsduhweb.rti.org/respweb/confidentiality.html.

An exempt IRB approval was requested from Michigan State University as data for this study was directly accessed from the RDAS online system. Thus, we did not collect any data, neither obtained consent from participants as well only had access to de-identified. data. The MSU IRB exempt approval number is IRB# 16-137/APP#i050607. This application was approved by the MSU IRB and determined to be Non Human Subject Research.

### Data Availability

This study was conducted with data from the National Surveys on Drug Use and Health data from 2002–2013, available to the public via Inter-university Consortium for

Political and Social Research websites (e.g., http://www.icpsr.umich.edu/icpsrweb/content/NAHDAP/about.html).

ICPSR also hosted the NSDUH online data analysis system known as the Restricted-Use Data Analysis System (R-DAS). R-DAS also can be used to produce our study estimates. R-DAS makes NSDUH data available via an online analysis interface designed to enable analysis-weighted contingency tables, but protects research participants by disabling data download features. For this reason, others can replicate our R-DAS analyses, but neither we nor others (outside government) have access to the confidential raw data.

The federal government recently announced its re-enablement of the R-DAS in a series of steps that is beginning with release of the P-DAS (Public-Use Data Analysis System). This is the URL link to the P-DAS, from which the future R-DAS URL links will be issued: https://www.datafiles.samhsa.gov/article/news/pdas-ready-use-nid16890.

### Supplemental Information

Supplemental information for this article can be found online at http://dx.doi.org/10.7717/peerj.2905#supplemental-information.

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
