# Peer review of "Epidemiological research on parent–child conflict in the United States: subgroup variations by place of birth and ethnicity, 2002–2013"

_PeerJ, doi:10.7717/peerj.2905_

## Round 0.1 · original submission · Major Revisions

· Academic Editor

Major Revisions

This is an interesting study focusing on the prevalence of parent-youth conflict in various age groups and ethnic groups. Despite its potential, the manuscript seems to have some concerns to address before being accepted for publication.

In addition to the comments from the other reviewer, please also consider the following 'review level' comments from myself.

Major concerns:
1) Sample selection bias. The latest US census showed that 8.5% of US population is naturalized US citizens and a similar portion of the population is non-US citizen (of course, foreign-born). [http://www2.census.gov/programs-surveys/demo/tables/foreign-born/2013/cps2013/2013-asec-tables-nativity.pdf] However, the included population had different portion/percentages of the US born and Foreign born subjects (e.g. 5% of non-Hispanic foreign whites vs about 15% overall foreign born reported in the Census data). This is a potential limitation.
2) Since the authors studied the frequency/prevalence of the PC conflicts in a time period of more than 10 years, it may be considered to compare the changes in the trends (or difference in the difference) of various groups.
3) It is known that race is linked to foreign born status and socioeconomic status. Therefore, income-levels of the household, female-employment status and probably should be included in the multivariate analyses (adjusted-OR computing).

Minor concerns:
1) Abstract has over 300 words, and should be shortened according to the PeerJ instruction. It should in my view show the numerical data including 95% confidence intervals and p values.
2) It is agreed that parent-children may be better fit for the manuscript than parent-youth.
3) Table 1. Only about 15-27%v of the non-Hispanic Asians are Chinese-Asian, who share much cultural and moral characteristics with other Koreans, Japanese and Singaporeans. What is the rationale of separating Chinse from other Asians? Please provide the sum of each columns. Two columns showing all US born and all foreign born subjects, respectively, are needed.
4) Table 2, ethnic self-identification groups in the title probably should be replaced with self-identified ethnic groups.
5) Some grammar and writing style issues (e.g. to sum up in my humble opinion should be in summary, in conclusion or taken together). Careful proof-reading is needed.

Reviewer 1 ·

Basic reporting

No Comments

Experimental design

No Comments

Validity of the findings

No Comments

Additional comments

This is an interesting epidemiologic study on the parent-children conflicts and tis associated factors. The author investigated whether the US born status is link to these conflicts in a multivariate model. Some concerns are raised for the authors’ consideration.

Major points:
1. Actually, the study of parent-child conflict is a very interesting subject whether in the United
States or across the globe, and the results of the paper are convincing to show us the epidemiological characteristics of parent-child conflict. However, the aim of the study is unclear and the related introduction and discussion could be expanded to further justify the study and promote its sociological impact. The authors should explain why there is any significant difference between Subgroup variations by place of birth and ethnicity.
2. Some very important opinions listed in the discussion section need further exploration. For
example, as a sociological paper, the reasons of the parent-youth conflict in the United States between Subgroup variations by the place of birth and ethnicity should be analyzed in more detail. Moreover, the corresponding intervention measures may be considered to mitigate the conflict.
3. Table 1 should have shown the numbers and percentages of children with reported
parent-child conflict of each group.
4. What was the reference group in the race/ethnicity variable in Table 2?
5. The authors may also consider include additional factors such as the socioeconomic status
of the family, the work status of the mother, and education levels of the parents.
6. The authors’ data cover an 11-year time frame. Were there any changes in the incidence and
its trends of Parent-child conflict in various ethnicity groups?
7. Some writing style and grammatical errors are present.

Minor points
1. Some references could be updated.
2. Parent-child conflict seems more appropriate than parent-youth. (see Journal of Family
Issues 29(6):780)
3. The manuscript also needs meticulous review and check, especially those key numbers in
the results. Obviously, there is a minor error about the number used in results line 311.
4. Table 2 title: frequent parent-youth conflict probably should be replaced with frequency of
parent-child conflict. Moreover, because only 3 additional covariates were included, they probably should have been reported.

---

## Round 0.2 · accepted · Accept

· Academic Editor

Accept

I have considered your responses and revised manuscript and feel that the article is now Acceptable. Thanks for your understanding and patience!